# Postprandial Abdominal Pain Caused by Gastroptosis—A Case Report

**DOI:** 10.3390/children10010116

**Published:** 2023-01-05

**Authors:** Anna Staszewska, Anna Jarzumbek, Anna Saran, Sylwia Gierak-Firszt, Jaroslaw Kwiecien

**Affiliations:** 1Department of Pediatrics, Institute of Medical Sciences, University of Opole, 47-100 Strzelce Opolskie, Poland; 2Department of Pediatrics, Faculty of Medical Sciences in Zabrze, Medical University of Silesia in Katowice, 41-800 Zabrze, Poland; 3Department of Radiology and Radiodiagnostics, Faculty of Medical Sciences in Zabrze, Medical University of Silesia in Katowice, 41-800 Zabrze, Poland; 4Szpital pod Bukami, 43-300 Bielsko-Biala, Poland

**Keywords:** gastroptosis, Glenard’s disease, abdominal pain

## Abstract

Gastroptosis is a condition in which the stomach is displaced downward and is a condition affects the spontaneous muscle mobility in the stomach. The reason for its current prevalence remains unclear as the medical literature is scarce on the condition in children. In this study, we describe the case of a 17-year-old girl suffering from chronic, position-dependent epigastric pain. The symptoms were observed during post-meal activity, with a significant increase in pain intensity while in an upright position. An inferior stomach displacement was noted in an upper gastrointestinal X-ray study using barium.

## 1. Introduction

Chronic abdominal pain in children is common, with up to a third of elementary students reporting abdominal pain weekly [1,2]. Functional gastrointestinal disorders, such as irritable bowel syndrome and functional dyspepsia, are the most common causes of chronic abdominal pain [1]. Other conditions, such as inflammatory bowel disease or allergic gut disorders, may also be the cause of such complaints.

Gastroptosis is a disorder with a downward displacement of the stomach, where its greater curve is dislocated below the level of the iliac crest. The diagnosis can be made with the use of imaging studies of the abdomen in an upright position [1]. Epigastric pain, abdominal discomfort, early satiety, loss of appetite, or even gastric emptying disorders can be caused by gastroptosis.

Symptoms are usually exacerbated during postprandial activity and differential diagnosis usually includes gastritis, pancreatitis, and peptic ulcer disease [1,2]. However, certain anatomic conditions, such as congenital stenosis of the duodenum and intestines, annular pancreas, pancreatic tumors, or intestinal malrotation, should also be considered. This would also include Crohn’s disease affecting the upper gastrointestinal (GI) tract, gastroesophageal reflux disease (GERD), food allergies, or psychosomatic disorders [3]. The symptoms caused by gastroptosis can impact the quality of life due to their postprandial character, which can disrupt daily activities [3,4].

## 2. Case Presentation

A 17-year-old female patient was admitted to the Children’s Gastroenterology Division for a planned diagnostic stay due to chronic abdominal pain.

### 2.1. Family History

Her family history was uneventful. The child was born of a single pregnancy, natural childbirth at 38th week of gestation with a birth weight of 2850 g. She was breastfed until the first month of age and her development was harmonious.

### 2.2. Medical History

At five years of age, the patient contracted varicella. She was diagnosed with Graves’ disease at 16 after exhibiting symptoms of hyperthyroidism: exophthalmos and hyperhidrosis. The diagnosis was confirmed in laboratory testing with a TSH level < 0.005 μIU/mL (N 0.27–4.2 μIU/mL), fT4 2.54 ng/dL (N 0.93–1.7 ng/dL), and TRAb 6.96 IU/L (N 0.3–1.58 IU/L). Her body weight was 50 kg (10–25 percentile) with a height of 165 cm (50th percentile) resulting in a BMI of 18.4 kg/m^2^ (10th percentile) when the diagnosis was made. She subsequently underwent thiamazole therapy which resulted in a state of euthyreosis and withdrawal of the aforementioned symptoms. The patient remained under the care of an allergist due to atopic dermatitis and an allergic response to numerous inhalatory allergens. An allergic reaction to trimethoprim-sulfamethoxazole was observed.

### 2.3. Symptoms

The onset of her symptoms occurred two years prior in a gradually increasing manner. The patient’s chief complaint was epigastric pain which was exacerbated after meals. There appeared to be a significant increase in its intensity in the upright position. The patient had no prior history of diarrhea or constipation. Under the care of her general practitioner (GP) she underwent basic laboratory tests with no significant findings, and an additional abdominal ultrasound found no abnormalities. However, a noteworthy weight loss was observed, with the patient losing 6 kg in body weight over the course of the last 6 months before admission. This may have been due to a change of eating habits and a new diet with increased dietary fiber consumption. The patient underwent regular thyroid screening and maintained a state of euthyreosis. Prior to admission, a gastroscopy was performed on an outpatient basis. Both endoscopic and microscopic examinations revealed a normal image of the stomach and duodenal mucosa.

### 2.4. Physical Examination, Laboratory Tests’ Results and Imaging Studies

Notable clinical findings on admission included significant decrease in body weight of 44 kg resulting in a BMI of 16 kg/m^2^ (both below the third percentile) and scoliosis. Laboratory tests on admission did not reveal any abnormalities with a TSH of 4.2 μIU/mL (N 0.27–4.2 μIU/mL), fT4 1.06 ng/dL (N 0.93–1.7 ng/dL), total IgA 1.13 g/L (N 0.61–3.48), and anti-tissue transglutaminase antibodies IgA 1.38 RU/mL (N < 20 RU/mL). Fecal calprotectin (FC) levels were also normal (<100 μg/g, N < 100 μg/g), which ruled out inflammatory bowel disease with a high degree of probability. Initial imaging studies, which included an abdominal ultrasound and a chest X-ray, did not reveal significant abnormalities other than the aforementioned scoliosis. During hospitalization, any eating disorders were excluded after an assessment by a trained clinical psychologist. However, it was noted that the patient exhibited symptoms that were indicative of depression and further psychological care was recommended.

As a result of initial diagnostics being inconclusive, an upper GI X-ray study using barium was performed, revealing a downward displacement of the stomach. In the standing position, the organ was located entirely under the diaphragm and extended all the way to the pelvic cavity (Figure 1 and Figure 2). During the procedure gastric emptying abnormalities were also observed, with no barium reaching the duodenum. However, in the supine position, contrast medium flow was unimpeded with both the duodenum and small intestine seemingly devoid of pathologies (Figure 3).

### 2.5. Diagnosis and Treatment

Considering the patient’s symptoms and imaging studies’ results, she was diagnosed with gastroptosis accompanied by abnormal gastric emptying exacerbated by weight loss. Since the patient had a prolonged QTc interval, cisapride therapy was ruled out and itopride was used instead. An adequate meal plan was suggested, aiming for weight gain, split into 5 meals eaten slowly. The patient was released home but remained in outpatient care.

### 2.6. Follow-Up

The patient was once again admitted to the hospital 6 months after initial discharge home due to a recurrence of weight loss and epigastric pain. On admission, her weight was 36.7 kg (<3 percentile), and with a height of 165.9 cm (50th percentile) her BMI was 13.3 kg/m^2^ (<3 percentile). Furthermore, amenorrhea was observed. An extensive laboratory testing panel revealed signs of hypogonadotropic hypogonadism with a low estradiol level, elevated cortisol, and hyperprolactinemia. A thorough psychiatric assessment of patient resulted in the diagnosis of anorexia nervosa (AN). Psychotherapy and anti-depression treatment were implemented alongside a high-calorie diet and a resting lifestyle. However, epigastric pain still persisted even after a weight gain of 4.2 kg. Further investigation of the patient revealed that she had not been taking the prescribed prokinetic drugs. She currently remains in outpatient care with a stable weight.

## 3. Discussion

Visceroptosis, the displacement of abdominal organs below their natural position is also known as Glenard’s disease. When the stomach is affected, the disease is referred to as gastroptosis. Gastroptosis is diagnosed when the stomach is observed to be displaced downwards, with its greater curve being partly projected below the level of the iliac crest, whilst the antrum remains in its usual position [1]. The first report of gastroptosis was published by Glenard in 1833. Since then, only a handful of cases were reported in the medical literature. Available data to date shows that gastroptosis mostly affects women between 20 and 50 years of age, with risk factors of low weight and postural defects being noted [4,5,6,7,8,9]. It is most likely caused by excessive laxity of the abdominal wall, with the mesenteric attachments of the stomach becoming too thin and relaxed under the organ’s weight. Additionally, a decreased amount of visceral fat within the lesser omentum is another predisposing factor [3,6]. Although gastroptosis has no specific symptoms, the most common manifestations of this condition have been noted to include epigastric pain or discomfort, nausea, emetic episodes, early satiety, and flatulence, which are all exacerbated in the standing position and during post-meal activity [2,6,9,10]. Other rare complications, such as abdominal circumference enlargement, ileus, and femoral hernia, have also been recorded [9,11]. Kusano et al. suggests that patients suffering from gastroptosis are less prone to dyspepsia [8]. However, such findings have not been reported in western populations. Upper GI tract fluoroscopy alongside oral barium or iodine-based contrast remains the main diagnostic tool, with additional high-resolution sequential X-rays in the standing position further helping diagnosis [1,2,4]. Gastroptosis is diagnosed when the stomach is displaced downwards, with its greater curve being partly projected below the level of the iliac crest. The antrum remains in its usual position [1]. As exemplified by our case, the condition is often accompanied by gastric emptying disorders, known as gastroparesis [4,12]. In children, the most common causes of gastroparesis include diabetes, viral infections, hypothyroidism, muscular dystrophy, and immaturity of the gastrointestinal tract in preterm babies [13,14]. Laboratory or endoscopic findings are not enough for the identification of gastroparesis. Differential diagnosis should include GERD, celiac disease, inflammatory bowel diseases, and various disorders of the pancreas and gallbladder. Furthermore, gastric emptying disorders may accompany gastroptosis, and for that reason, bowel obstruction should be also taken under consideration. In our case, the patient did experience early satiety. In Glenard’s times, the usual approach was to perform surgery [15]. Nowadays, invasive treatment is reserved to a handful of cases complicated by bowel obstruction [4,10,16,17]. Contemporary treatment includes prokinetic drugs and an adequate diet. Physiotherapy focused on abdominal and paraspinal muscles’ strengthening also plays an important role. At times, special abdominal bands are used [2,10,16,17].

## 4. Summary and Conclusions

Although our patient’s medical history revealed risk factors for gastroptosis, including female sex, weight deficiency, and a postural defect (scoliosis), diagnosis of gastroptosis was prolonged due to weight loss not being reported in the current medical literature. As such, the aforementioned weight loss was seen to be one of the causing factors for gastroptosis in the described case. The diagnosis was made furthermore difficult due to the patient’s medical history which included hyperthyroidism (Graves-Basedow disease) and depression. She was also diagnosed with anorexia nervosa a few months later during her second hospital stay. Gastroptosis was confirmed in an upper GI tract X-ray using barium. As the aforementioned diagnostic method is both easily available and interpreted, it should be considered as an imaging modality when gastroptosis is suspected. It is the authors suggestion that gastroptosis and secondary gastrointestinal tract motility disorders should be taken into consideration when diagnosing non-specific dyspeptic symptoms in children.

## Figures and Tables

**Figure 1 children-10-00116-f001:**
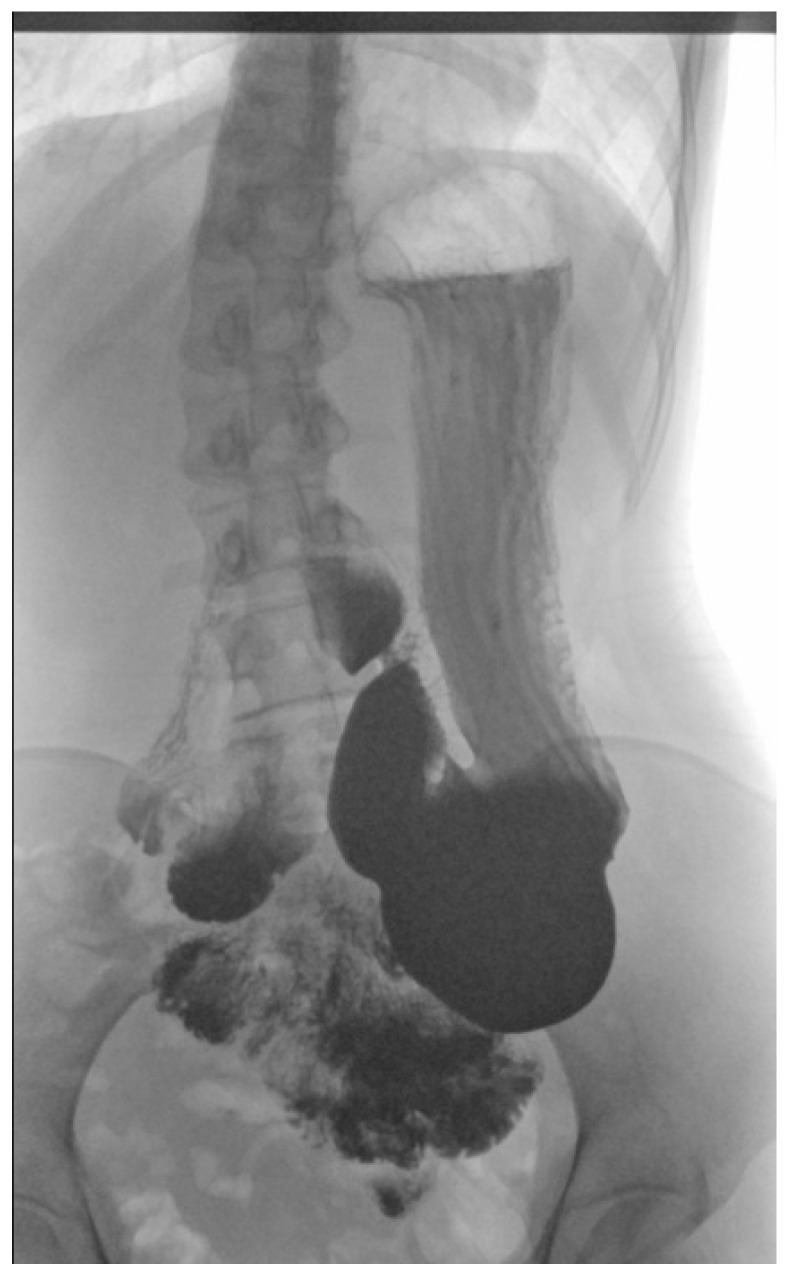
The stomach and duodenum clearly visible in an upper GI X-ray using barium in upright position. The greater curve is reaching downward to the sacrum. Please note the retention of barium contrast in the stomach.

**Figure 2 children-10-00116-f002:**
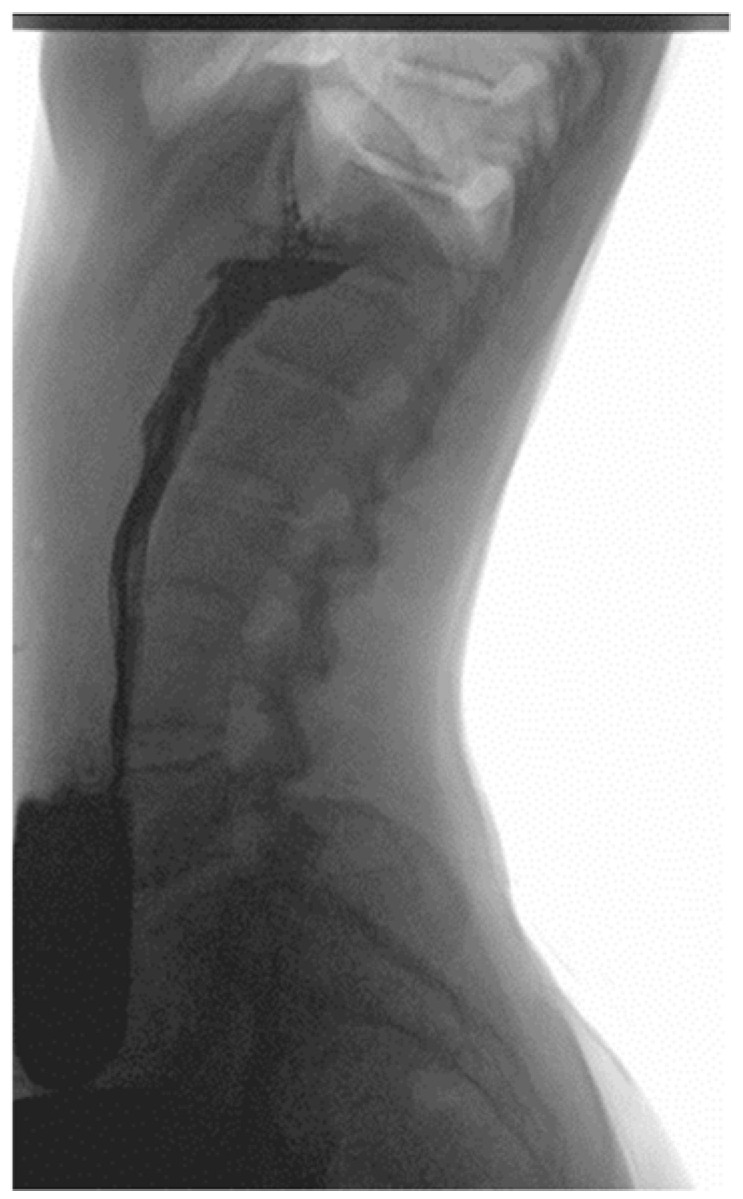
The greater curve of the stomach is displaced downward, below the level of the iliac crest. Lateral view in upright position.

**Figure 3 children-10-00116-f003:**
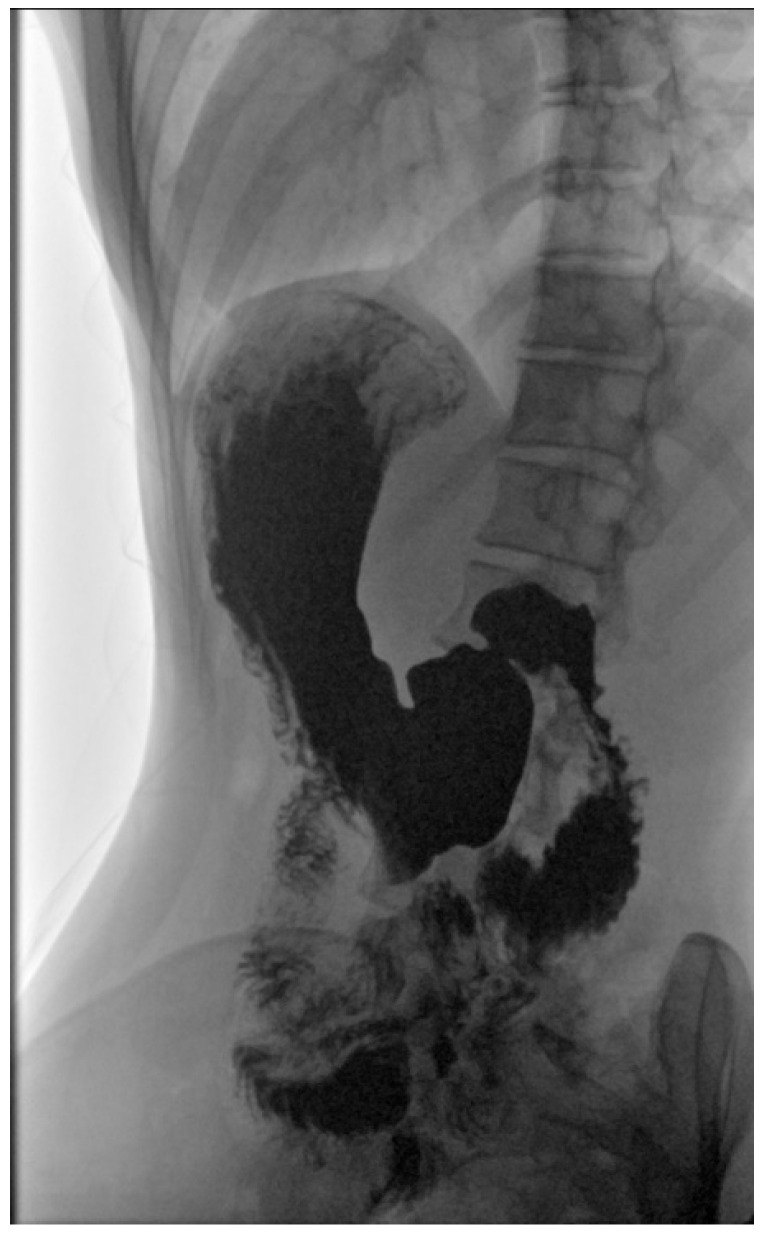
Efficient barium contrast emptying in supine position. The duodenum and intestines are devoid of pathologies.

## Data Availability

Not applicable.

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
