# Peer review of "Postprandial Abdominal Pain Caused by Gastroptosis—A Case Report"

_children, 2023, doi:10.3390/children10010116_

Round 1

Reviewer 1 Report

I have reviewed the case report paper entitled: “A Postprandial Abdominal Pain Caused by Gastroptosis – A Case Report”. In this paper, the authors describe the clinical case of a sporadic disease called Gastroptosis or formerly known as Glenard’s disease in a 17 years-old female. I consider that the introduction is correct since it provides information about the disease and the part of the clinical description of the patient is right. The images were well taken and the discussion was very complete.

I believe that they should be more specific with the laboratory and imaging tests that they used for diagnosis.

I think that the captions could include a more detailed description of the images.

Author Response

First of all, we want to sincerely express our thanks for the constructive and precise feedback of our paper. We agree with the reviewers’ notes and caveats and have given our best effort to follow them through. All of the suggested changes were addressed, including a language check by the native speaker with medical education and we look forward to further evaluation. What follows are our answers to each of the reviewers.

Peer review no. 1

  1. “I believe that they should be more specific with the laboratory and imaging tests that they used for diagnosis.”

We included additional lab results and modified the descriptions of imaging modalities used.

  1. “I think that the captions could include a more detailed description of the images.”
    Issue addressed. All the captions were rewritten to be more precise.

Reviewer 2 Report

Overall an interesting report but the manuscript lacks a logical flow of the case and sufficient information.

Introduction.

Give some examples of structural abnormalities that may cause similar symptoms/differential diagnoses, and what impact they may have. May affect quality of life, etc if evident in other conditions?

Case report

This section needs to be presented more logically in terms of timeline. Use sub-headings if helpful.

Add additional information on what other conditions were tested for and how excluded? 2 years is a long time to have no previous assessments carried out so please expand this section..

For variables such as weight, for example – what BMI z score etc she was at start of abdo pain 2 years prior, and change over time – was it only in last 6m she dropped weight? Was hyperthyroidism tested for cause of weight loss? Did she start on meds for this etc.

You need more information on the gastric emptying results – time to empty etc as this is wholly relevant for clinicians to compare and contrast.

The paragraph about eating disorders should surely come before the radiology?

There needs to be additional information about follow-up. Response to itoprode, symptom resolution, weight gain/loss, etc, any follow-up scans?

Discussion

This seems a bit empty. Additional comparison of your patient with other literature, if available, would be helpful… time to diagnosis, long-term outcomes etc?

Author Response

First of all, we want to sincerely express our thanks for the constructive and precise feedback of our paper. We agree with the reviewers’ notes and caveats and have given our best effort to follow them through. All of the suggested changes were addressed, including a language check by native speaker with medical education and we look forward to further evaluation. What follows are our answers to each of the reviewers.

Peer review no. 2                                                                              

  1. „Overall an interesting report but the manuscript lacks a logical flow of the case and sufficient information.”

The report was split into subsections and its order modified. We have added more detail to the report.

  1. „Give some examples of structural abnormalities that may cause similar symptoms/differential diagnoses, and what impact they may have. May affect quality of life, etc if evident in other conditions?”

We now discuss the differential diagnosis options on more detail and have referred to this.

  1. „Case report. This section needs to be presented more logically in terms of timeline. Use sub-headings if helpful.”

The entire section was reworked.

  1. „Add additional information on what other conditions were tested for and how excluded? 2 years is a long time to have no previous assessments carried out so please expand this section.”

We have expanded the section following the advice.

  1. „For variables such as weight, for example – what BMI z score etc she was at start of abdo pain 2 years prior, and change over time – was it only in last 6m she dropped weight? Was hyperthyroidism tested for cause of weight loss? Did she start on meds for this etc.”

Missing anthropometric measurements were added as suggested. Hyperthyroidism was considered and ruled out as the cause for weight loss.

  1. „You need more information on the gastric emptying results – time to empty etc as this is wholly relevant for clinicians to compare and contrast.”

Contrast medium was present in the duodenum 11 minutes after being found in the stomach.

  1. „The paragraph about eating disorders should surely come before the radiology?”

We changed the article’s structure according to the suggestions.

  1. „There needs to be additional information about follow-up. Response to itoprode, symptom resolution, weight gain/loss, etc, any follow-up scans?”

The section was expanded to include more information on the follow-up.

  1. „This seems a bit empty. Additional comparison of your patient with other literature, if available, would be helpful… time to diagnosis, long-term outcomes etc?”

We modified the section to be more detailed. However, it is worth noting how scarce the literature on the topic is; very little actual data is available to us in this case.

Reviewer 3 Report

This MS describes a case of one adolescent diagnosed with gastroptosis

SPECIFIC COMMENTS

1. The summary refers to gastric elongation, but the reports focuses on gastroptosis: these terms don't have the same meaning

2. The INTRO is appropriately short and focused. However, most of the sentences in the INTRO are awkward or poorly worded: extensive revision required. The first sentences (for example) should read as: "Chronic abdominal pain in children is common, with up to a third of elementary students reporting pain weekly [1,2]. Functional gastrointestinal disorders, such as irritable bowel syndrome and functional dyspepsia, are the most common causes of chronic abdominal pain [1]. Pain can also occur in other conditions, such as inflammatory bowel disease or allergic gut disorders.

3. at line 35: is this days, months or years?

4. line 36: "with laboratory testing"

5. Line 39: unburdened is used incorrectly here. 

6. line 47: the word macroscopic should only be used to describe the inspection by naked eye. Here the word macroscopic should be replaced with the word endoscopic, reflecting that this was the assessment at that time

7. line 49: Patient should be patient

8. Line 49: what is meant by "quite good"? Perhaps comfortable and undistressed??

9. helpful to list the actual BMI as well as the location on chart. It was not clear if this BMI reflected a loss of weight? If so, include the extent of weight loss

10.  line 51:  "did not reveal anything out of the norm" is a little informal for a scientific publication

11. line 56: was this a barium meal and follow-through OR a barium meal? A suitable description could also be "upper gastrointestinal contrast study using barium"

12. lines 59/60: this suggests complete obstruction. Later the term gastroparesis was employed. Did the patient actually have symptoms consistent with poor gastric clearance?

13. line 70: "exhibited depressive symptoms"

14. line 76: the phrase "the patient remains under control" is an unusual phrase. Perhaps this should state that the patient's symptoms had resolved?

15. abbreviations should be explained in full prior to use (e.g. GERD)

16. iflammatory needs correction

17. The phrase XIX century cold be rewritten

18. line 113: needs to be rewritten

19. lines 114/115: the word burdened is not standard here. Perhaps this sentence should summarise what risk factors the patient actually had

20. line 116: "upper gastrointestinal tract contrast study"

21. consider using the abbreviation GI when gastrointestinal term is first used

22. Please review all references to ensure that all fully comply with the journal requirements. Reference #7 is one example that needs correction

23. A number of other sentences with awkward or inappropriate language or word usage that also need correction

Author Response

First of all, we want to sincerely express our thanks for the constructive and precise feedback of our paper. We agree with the reviewers’ notes and caveats and have given our best effort to follow them through. All of the suggested changes were addressed, including a language check by native speaker with medical education and we look forward to further evaluation. What follows are our answers to each of the reviewers.

Peer review no. 3                                                                  

  1. The summary refers to gastric elongation, but the reports focuses on gastroptosis: these terms don't have the same meaning

We modified the text to account for that error.

  1. The INTRO is appropriately short and focused. However, most of the sentences in the INTRO are awkward or poorly worded: extensive revision required. The first sentences (for example) should read as: "Chronic abdominal pain in children is common, with up to a third of elementary students reporting pain weekly [1,2]. Functional gastrointestinal disorders, such as irritable bowel syndrome and functional dyspepsia, are the most common causes of chronic abdominal pain [1]. Pain can also occur in other conditions, such as inflammatory bowel disease or allergic gut disorders.

The entire section was rewritten.

  1. at line 35: is this days, months or years?

Modified

  1. line 36: "with laboratory testing"

Modified

  1. Line 39: unburdened is used incorrectly here. 

Modified

  1. line 47: the word macroscopic should only be used to describe the inspection by naked eye. Here the word macroscopic should be replaced with the word endoscopic, reflecting that this was the assessment at that time

Modified

  1. line 49: Patient should be patient

Modified

  1. Line 49: what is meant by "quite good"? Perhaps comfortable and undistressed??

Modified

  1. helpful to list the actual BMI as well as the location on chart. It was not clear if this BMI reflected a loss of weight? If so, include the extent of weight loss

Done

  1. line 51:  "did not reveal anything out of the norm" is a little informal for a scientific publication

Modified

  1. line 56: was this a barium meal and follow-through OR a barium meal? A suitable description could also be "upper gastrointestinal contrast study using barium"

Modified

  1. lines 59/60: this suggests complete obstruction. Later the term gastroparesis was employed. Did the patient actually have symptoms consistent with poor gastric clearance?

We added more details on that. Yes, the patient reported early satiety, an information that was omitted in our manuscript. The section of the text has been reworked.

  1. line 70: "exhibited depressive symptoms"

Modified

  1. line 76: the phrase "the patient remains under control" is an unusual phrase. Perhaps this should state that the patient's symptoms had resolved?

Modified

  1. abbreviations should be explained in full prior to use (e.g. GERD)

Done

  1. iflammatory needs correction

Corrected

  1. The phrase XIX century cold be rewritten

Done

  1. line 113: needs to be rewritten

Done

  1. lines 114/115: the word burdened is not standard here. Perhaps this sentence should summarise what risk factors the patient actually had

Modified

  1. line 116: "upper gastrointestinal tract contrast study"

Modified

  1. consider using the abbreviation GI when gastrointestinal term is first used

Done

  1. Please review all references to ensure that all fully comply with the journal requirements. Reference #7 is one example that needs correction

This issue has been fixed

  1. A number of other sentences with awkward or inappropriate language or word usage that also need correction

The issue was resolved; some changes on the final version of the manuscript were mistakenly done after a language check.

Round 2

Reviewer 1 Report

The author corrected reviewers´ comments 

Author Response

Dear Editor,

Thank you for your review, we are happy that you have no other comments. We upload the version revised according to other Reviewer's suggestions.

Regards,
Jaroslaw Kwiecień

Reviewer 2 Report

This is vastly improved. The time taken to address my comments so thoroughly is appreciated.  As such I have no further edits. Well done!

Author Response

(The authors gave the same response as above.)

Reviewer 3 Report

Thank you for revisions. The MS is improved.

some minor issues are still present:

1. Suggest to break the INTRO into smaller paragraphs (rather than one long paragraph). The first break could be after the third sentence, for example.

2. there are some typographical errors. As one example "quality of life quality" in the INTRO needs correction

3. Line 96: sentences should start with words only and not with numerals. Six months....

4. line 91: this suggests that there were multiple disorders of gastric emptying present. Suggest to rephrase as "accompanied by abnormal gastric emptying" or "accompanied by disordered gastric emptying" 

5. this sentence (lines 171/172) is incomplete: 'As the aforementioned diagnostic method is both easily available and interpreted."

There are also some other awkward sentences that could also be improved with revision

Author Response

Dear Reviewer,

Thank you for your second review, I apologize for the delay caused by winter holidays of our native speaker consultant.

In the third revised version we fixed mistakes according to your suggestions, and the consultant being native speaker has been checked the manuscript one more time. We hope that the current version will be suitable for publication.

Your review point by point:

1. Suggest to break the INTRO into smaller paragraphs (rather than one long paragraph). The first break could be after the third sentence, for example.

We have corrected the text as suggested.

2. there are some typographical errors. As one example "quality of life quality" in the INTRO needs correction

I apologize for these mistakes in the second version. We have corrected the text as suggested.

3. Line 96: sentences should start with words only and not with numerals. Six months....

We have corrected the text as suggested.

4. line 91: this suggests that there were multiple disorders of gastric emptying present. Suggest to rephrase as "accompanied by abnormal gastric emptying" or "accompanied by disordered gastric emptying" 

We have corrected the text as suggested.

5. this sentence (lines 171/172) is incomplete: 'As the aforementioned diagnostic method is both easily available and interpreted."

I apologize for these mistakes in the second version. We have corrected the text as suggested.

There are also some other awkward sentences that could also be improved with revision.

I apologize for these mistakes in the second version. The independent native speaker (not being one of the authors) has made for us another review and corrections was inserted.

Regards,
Jaroslaw Kwiecień